# A Comparison of the Effectiveness of the Serratus Anterior Plane Block and Erector Spinae Plane Block to that of the Paravertebral Block in the Surgical Treatment of Breast Cancer—A Randomized, Prospective, Single-Blinded Study

**DOI:** 10.3390/jcm13164836

**Published:** 2024-08-16

**Authors:** Michał But, Krzysztof Wernicki, Jacek Zieliński, Weronika Szczecińska

**Affiliations:** 1Pain Treatment Clinic, Polyclinic in Koszalin, 75-720 Koszalin, Poland; 2The Department of Anesthesiology and Intensive Care, Provincial Hospital in Koszalin, 75-581 Koszalin, Poland; contact@krzysztofwernicki.pl; 3The Department of Surgical Oncology, Provincial Hospital in Słupsk, 76-200 Słupsk, Poland; jaziel@gumed.edu.pl; 4The Department of General Surgery, Hospital Copernicus in Gdańsk, 80-803 Gdańsk, Poland; ik5men@wp.pl

**Keywords:** regional anesthesia, breast surgery, paravertebral block, erector spinae plane block, serratus anterior plane block

## Abstract

**Background/Objectives:** The paravertebral block (PVB) is a well-studied, effective method of analgesia for breast surgery. Alternative techniques involving the blockage of intercostal nerve branches are the serratus anterior plane block (SAPB) and the erector spinae plane block (ESPB). However, no studies comparing both fascial blocks to PVB in breast surgery have been published to date. We evaluated the effectiveness of ESPB and SAPB vs. PVB, expressed as the requirement for intraoperative fentanyl, pain intensity at rest and during coughing, and morphine consumption on the first postoperative day. Additional aims were to perform an evaluation of the safety of the block types used. **Materials and Methods**: A total of 77 women and 1 man with stage I and II clinical breast cancer, aged 18–85 years, were randomized into one of three study groups: SAPB, PVB, and ESPB. **Results:** There were no statistically significant differences in fentanyl consumption during surgery with respect to the type of block used (*p* = 0.4246). Morphine consumption in the postoperative period was highest in the ESPB group, averaging 9.4 mg. There was a statistically significant difference in pain intensity from 4 pm on the day of surgery to 8 am the following morning. No complications related to the blocks were observed on the first postoperative day. **Conclusions:** Both the serratus anterior plane block and the erector spinae plane block were as effective as the paravertebral block in achieving intraoperative analgesia. The serratus anterior plane block was equally as effective as the paravertebral block in achieving postoperative analgesia. The erector spinae plane block was significantly less effective in achieving postoperative analgesia than both the paravertebral block and serratus anterior plane block.

## 1. Introduction

Breast cancer is one of the most frequently diagnosed cancers in the adult population worldwide (over 2.2 million new cases in 2022) [1], as well as in Poland (over 21,000 new cases in 2021) [2]. The standard procedure is surgery, the extent of which is defined by the stage of the disease. Surgical procedures include breast-conserving treatment (BCT), mastectomy, and axillary lymphadenectomy—removal of the sentinel node or all nodes in the axillary region. The specific and complex innervation of the surgical area originating from both the brachial plexus and intercostal nerves causes these procedures to be associated with significant nociceptive stimulation, resulting in severe postoperative pain and a high risk of persistent postoperative pain in 25 to 60% of cases [3] (Figure 1).

A well-documented and effective technique for disrupting intercostal nociception is the paravertebral block (PVB) [4]. Unfortunately, due to the anatomy of the paravertebral space, it carries a risk of pleural puncture and pneumothorax [5], as well as other potentially serious complications [6]. With the dynamic development of regional anesthesia, many new promising fascial blocks have emerged recently which can positively affect postoperative pain intensity and opioid consumption during and after surgery. Alternative blocks involving the anesthesia of intercostal nerve branches include the serratus anterior plane block (SAPB) and the erector spinae plane block (ESPB) [7,8,9,10]. There are many studies in the literature comparing the effectiveness of PVB to that of ESPB or SAPB. So far, there are no studies comparing the effectiveness of both fascial blocks to the paravertebral block. In accordance with PROSPECT recommendations, the researchers of this study decided not to create a control group in which no method of regional anesthesia was used. The control group included patients who underwent a PVB. 

## 2. Materials and Methods

The main objective of this study was to compare the efficacy of the fascial blocks ESPB (erector spinae plane block) and SAPB (serratus anterior plane block) to that of PVB (paravertebral block) in breast surgery.

The specific objectives of this study were as follows:To assess the impact of SAPB, ESPB, and PVBs on fentanyl consumption during surgery.To assess the impact of SAPB, ESPB, and PVBs on the amount of morphine administered to patients postoperatively.To assess the impact of SAPB, ESPB, and PVBs on pain intensity during coughing and at rest on the first day post-surgery six times daily from the first hour post-operation until the end of the second postoperative day.To evaluate the safety of the paravertebral block and the applied fascial blocks SAPB and ESPB.

After obtaining written consent for the study, patient numbers were assigned, and the block was performed according to a randomization table. For all blocks, patients were positioned on the side opposite to the operative side. All regional blocks were performed under ultrasound guidance (Mindray TE7) using a linear probe by a single experienced anesthesiologist. For the blocks, 20 mL of local anesthetic (10 mL of 2% lidocaine with 10 mL of 0.5% bupivacaine) was used.

This study included patients diagnosed with breast cancer at clinical stages I and II, aged between 18 and 85 years. Patients excluded from the study were those who were unable to provide informed consent, had undergone previous breast surgery on the same side, reported chronic pain with an intensity of NRS > 3, had been taking opioids chronically up to 2 weeks before the surgery, had contraindications to regional anesthesia (such as allergies to local anesthetics, with dermatological conditions preventing regional block), had a BMI > 40, or had consciousness disorders or difficulty with verbal communication (Figure 2). Patients were prospectively and randomly assigned to groups with a single-blind design. A computer-generated table in Microsoft Excel assigned patients to the appropriate groups.

### 2.1. Serratus Anterior Plane Block (SAPB)

The probe was placed on the mid-axillary line at the level of the fourth rib to visualize the serratus anterior and latissimus dorsi muscles (Figure 3). After establishing the correct level, the skin was anesthetized with 1 mL of 1% lidocaine, and a SonoPlex Pajunk 50 mm needle was introduced in-plane. After puncturing the serratus muscle and contacting the rib, 20 mL of local anesthetic was deposited between the serratus muscle fascia and the rib periosteum, termed a deep serratus anterior plane block.

### 2.2. Paravertebral Block (PVB)

The probe was placed between the fourth and fifth ribs. After establishing the correct plane, the skin was locally anesthetized with 1 mL of 1% lidocaine. The needle was introduced in-plane under the internal intercostal membrane (Figure 4). Following negative aspiration, 20 mL of local anesthetic was deposited, showing the retraction of the pleura.

### 2.3. Erector Spinae Plane Block (ESPB)

The probe was placed in the paravertebral area to visualize the transverse process and the fifth rib. After establishing the correct plane, the skin was anesthetized with 1 mL of 1% lidocaine. A SonoPlex Pajunk needle was introduced out-of-plane until contact with the periosteum. The probe was then rotated 90 degrees to visualize the needle throughout its course (Figure 5). Following negative aspiration, 20 mL of local anesthetic was deposited between the fascia of the erector spinae muscle and the periosteum of the transverse process.

### 2.4. Perioperative Management

After performing the block in the operating room, general balanced anesthesia was conducted following a standardized protocol. Co-induction included 2 mg of midazolam and 0.1 mg of intravenous fentanyl. Pre-oxygenation with 100% oxygen was followed by intravenous induction with propofol at a dose of 2 mg/kg of body weight. Muscle relaxation was achieved with rocuronium 0.6 mg/kg, dosed based on ideal body weight, and maintenance was carried out with inhaled desflurane. Additional doses of 0.1 mg fentanyl were administered if blood pressure exceeded 120% of the baseline values. Post-surgery, patients were monitored in the recovery room until fully conscious, and then connected to a PCA pump with morphine.

### 2.5. Postoperative Pain Intensity Measurement

During the one-hour observation period in the recovery room, vital signs and pain intensity were monitored using an 11-point Numerical Rating Scale (NRS). Patients were asked to rate their pain at rest and then to cough and rate their pain during coughing on a scale from 0 to 10, with 0 indicating no pain and 10 indicating the worst pain imaginable. The highest reported pain score during the first hour post-operation was recorded. Subsequent measurements were conducted by a nurse in the ward at scheduled intervals, assessing pain both at rest and during coughing in the same manner using the same NRS.

### 2.6. Complication Evaluation

Before transferring patients to the ward, the regional anesthesia site of the block was scanned using ultrasound by a skilled radiologist for adverse events such as hematoma or pneumothorax.

### 2.7. Statistical Analysis

All statistical calculations were performed using TIBCO Software Inc. (Palo Alto, CA, USA) (2017), Statistica (data analysis software system), version 13 and Microsoft Excel. Quantitative variables were characterized by the arithmetic mean, standard deviation, median, minimum and maximum values (range), and 95% confidence interval (CI). Qualitative variables are presented as frequencies and percentages.

### 2.8. Sample Size and Power of the Study

An ANOVA was conducted. An analysis of PCA morphine requirements on the first postoperative day was performed for three independent groups. Under the conditions of α = 0.05 and power = 80–90%, the minimum sample size obtained was 3–24. Assuming equal numbers in all the studied groups and a population standard deviation value at even a high level of 4–5, the minimum sample size for each group should be approximately 20–24 patients. 

As a second variable, pain intensity was analyzed, measured one hour after the procedure, and then every 6 h. Under the conditions of α = 0.05 and power = 80–90%, the minimum sample size obtained was 4–16. Assuming equal numbers in all studied groups and the number of pain/rest intensity measurements at the level of 8, the sample size for each group should be approximately 20 patients.

Additionally, a power analysis was conducted for the following calculations. The power obtained was 97% for both variables.

## 3. Results

The average ages of patients in the SAPB, PVB, and ESPB groups were 66.5, 66.5, and 62.5 years, respectively. No statistically significant differences in age were found between the block types (*p* = 0.4577). The average body weights of patients in the SPB, PVB, and ESPB groups were 74.1, 69.3, and 73.9 kg, respectively. No statistically significant differences in body weight were found between the block types (*p* = 0.2898). The average heights of patients in the SAPB, PVB, and ESPB groups were 159.5, 158.7, and 163.9 cm, respectively. Patients in group 2 were significantly shorter than those in group 3 (*p* = 0.0291). The average BMIs of patients in the SPB, PVB, and ESPB groups were 28.9, 27.4, and 27.8, respectively. No statistically significant differences in BMI were found between the block types (*p* = 0.7089).

A total of 78 procedures were performed. The majority were breast-conserving surgeries with sentinel lymph node excision, totaling 47. Twelve mastectomies with sentinel lymph node excision were performed, along with ten Maddens’ mastectomies, three tumorectomies, two BCTs (breast-conserving treatments), and one quadrantectomy. Detailed numbers and percentages of the procedures are presented in Table 1. There were no statistically significant differences in the types of procedures with respect to the type of block used (*p* = 0.8031).

The percentages of patients with ASA scale 1 in the SAPB, PVB, and ESPB groups were 4.2%, 3.4%, and 8.3%, respectively, and those of patients with ASA scale 2 were 95.8%, 96.6%, and 91.7%. No statistically significant differences in ASA scale were found between the block types (*p* = 0.7008). The average NRS pain scale scores of patients in the SAPB, PVB, and ESPB groups were 0.48, 0.41, and 0.29, respectively. No statistically significant differences in preoperative NRS pain scale scores were found between the block types (*p* = 0.7710).

### 3.1. Intraoperative Fentanyl Consumption and Postoperative Morphine Consumption

The average fentanyl consumption values during surgery for patients in the SAPB, PVB, and ESPB groups were 0.12 mg, 0.12 mg, and 0.11 mg, respectively. There were no statistically significant differences in fentanyl consumption during surgery with respect to the type of block used (*p* = 0.4246). The average postoperative morphine consumption values for patients in the SAPB, PVB, and ESPB groups were 5.4 mg, 4.4 mg, and 9.4 mg, respectively. In the ESPB group (erector spinae plane block), consumption was significantly higher compared to the SAPB group (*p* = 0.0074) and the PVB group (*p* = 0.0005) (Table 2).

### 3.2. Pain Intensity at Rest

The first measurement of pain intensity at rest was conducted in the recovery room. Subsequent measurements were carried out by a nurse in the patient’s room.

The average pain intensity at rest 1 h after the procedure for patients in the SAPB, PVB, and ESPB groups was 4.1, 4.0, and 3.9, respectively, according to the NRS. There were no statistically significant differences in pain intensity at rest 1 h after the procedure with respect to the type of block used (*p* = 0.8651).

The average pain intensities at 4:00 PM for patients in the SAPB, PVB, and ESPB groups were 2.6, 2.6, and 3.6, respectively, according to the NRS. At 4:00 PM, there were significant differences in pain levels at rest depending on the type of anesthesia block used (*p* = 0.0017). Detailed post hoc tests showed that in the ESPB group (erector spinae plane block), pain intensity was significantly higher compared to the SAPB group (*p* = 0.0129) and the PVB group (*p* = 0.0071). No significant differences were found for the other comparisons.

The average pain intensities at 8:00 PM for patients in the SAPB, PVB, and ESPB groups were 2.6, 2.6, and 3.6, respectively, according to the NRS. There were statistically significant differences in pain intensity at rest at 8:00 PM with respect to the type of block used (*p* = 0.0017). Detailed post hoc tests showed that in the ESPB group, pain intensity was significantly higher compared to the SAPB group (*p* = 0.0129) and the PVB group (*p* = 0.0071). No significant differences were found for the other comparisons.

For the times 12:00 AM and 4:00 AM, statistical tests could not be calculated due to small or absent sample sizes.

The average pain intensities at 8:00 AM the next day for patients in the SAPB, PVB, and ESPB groups were 2.1, 2.0, and 2.7, respectively, according to the NRS. There were statistically significant differences in pain intensity at rest at 8:00 AM with respect to the type of block used (*p* = 0.0176). Detailed post hoc tests showed that in the ESPB group, pain intensity was significantly higher compared to the PVB group (*p* = 0.0388). No significant differences were found for the other comparisons.

The average pain intensities at 12:00 PM the next day for patients in the SAPB, PVB, and ESPB groups were 1.8, 1.9, and 2.4, respectively, according to the NRS. There were no statistically significant differences in pain intensity at 12:00 PM with respect to the type of block used (*p* > 0.05).

The average pain intensities at 4:00 PM the next day for patients in the SAPB, PVB, and ESPB groups were 1.8, 1.8, and 2.2, respectively, according to the NRS. There were no statistically significant differences in pain intensity at 4:00 PM with respect to the type of block used (*p* = 0.2182) (Table 3).

### 3.3. Pain Intensity during Coughing

Similarly to the pain measurement at rest, the first pain intensity measurement was conducted in the recovery room. Subsequent measurements were carried out by nurses on the ward. During the measurement, patients were asked to cough.

The average pain intensities during coughing 1 h post-operation for patients in the SAPB, PVB, and ESPB groups were 4.8, 4.8, and 4.8, respectively, according to the NRS. No statistically significant differences in pain intensity during coughing were found 1 h post-operation among the types of block (*p* = 0.8809).

The average pain intensities at 4:00 PM for patients in the SAPB, PVB, and ESPB groups were 3.5, 3.5, and 4.5, respectively, according to the NRS. Statistically significant differences in pain intensity during coughing were found at 4:00 PM among the types of block (*p* = 0.0010). Detailed post hoc tests showed that in the ESPB group (erector spinae plane block), pain intensity was significantly higher compared to the SAPB group (*p* = 0.0080) and PVB group (*p* = 0.0045). No other statistically significant differences were observed.

The average pain intensities at 8:00 PM for patients in the SAPB, PVB, and ESPB groups were 3.5, 3.5, and 4.5, respectively, according to the NRS. Statistically significant differences in pain intensity during coughing were found at 8:00 PM among the types of block (*p* = 0.0010).

Detailed post hoc tests showed that in the ESPB group (erector spinae plane block), pain intensity was significantly higher compared to the SAPB group (*p* = 0.0080) and PVB group (*p* = 0.0045). No other statistically significant differences were observed.

For 12:00 AM and 4:00 AM, statistical tests could not be performed due to small sample sizes.

The average pain intensities at 8:00 AM the next day for patients in the SAPB, PVB, and ESPB groups were 3.1, 3.1, and 3.7, respectively, according to the NRS. Statistically significant differences in pain intensity during coughing were found at 8:00 AM among the types of block (*p* = 0.0128). Detailed post hoc tests showed that in the ESPB group (erector spinae plane block), pain intensity was significantly higher compared to the PVB group (*p* = 0.0478). No other statistically significant differences were observed.

The average pain intensities at 12:00 PM the next day for patients in the SAPB, PVB, and ESPB groups were 2.8, 2.8, and 3.4, respectively, according to the NRS. Statistically significant differences in pain intensity during coughing were found at 12:00 PM among the types of block (*p* = 0.0097). Detailed post hoc tests showed that in the ESPB group (erector spinae plane block), pain intensity was significantly higher compared to the PVB group (*p* = 0.0253). No other statistically significant differences were observed.

The average pain intensities during coughing at 4:00 PM the next day for patients in the SAPB, PVB, and ESPB groups were 2.8, 2.6, and 3.2, respectively, according to the NRS. No statistically significant differences in pain intensity during coughing were found at 4:00 PM among the types of block (*p* > 0.05) (Table 4).

## 4. Discussion

### 4.1. Intraoperative Analgesia

In this study, both the serratus anterior plane block and the erector spinae plane block were as effective as the paravertebral block in achieving intraoperative analgesia. Intraoperative fentanyl use was similar in all groups, which is in line with the study by Gabriel [11].

### 4.2. SAPB vs. PVB in Postoperative Analgesia

In our study, SAPB was equally as effective as PVB in achieving postoperative analgesia. The reported pain levels and opioid use were similar in both groups.

In the study by Gabriel et al., the serratus anterior plane block was reported to be less effective than the paravertebral block. The reported pain levels and postoperative opioid use were higher in the serratus plane block group compared to the paravertebral block group [11]. The discrepancies between their results and ours may be attributable to several factors. Gabriel et al. describe non-mastectomy procedures primarily performed on outpatients. Additionally, some blocks, particularly serratus anterior plane blocks, were performed by less experienced practitioners in training, which may have affected their effectiveness. Our study also includes radical procedures, exclusively in inpatients, with all blocks performed by a single experienced anesthesiologist.

In the study by Gupta et al., it was reported that the duration of analgesia from SAPB was significantly shorter compared to PVB. The reported pain levels and postoperative opioid use were higher in the SAPB group than in the PVB group. The authors speculated that the greater effectiveness of PVB is associated with a larger extent of the block [12].

Although the study by Gupta seems to lack the disadvantages of the study by Gabriel, both studies used the superficial serratus plane block, whereas in our study, we employed the deep serratus plane block. The more efficient option for mastectomy remains unclear [13,14,15].

### 4.3. ESPB vs. PVB in Postoperative Analgesia

In our study, ESPB was significantly less effective than PVB in achieving postoperative analgesia, which is consistent with findings from other similar studies [16,17,18] where opioid use and/or reported pain intensity were higher in the ESPB group compared to the PVB group. The greater effectiveness of PVB is likely associated with a larger extent of the block [19].

### 4.4. ESPB vs. SAPB in Postoperative Analgesia

In our study, ESPB was significantly less effective than SAPB in achieving postoperative analgesia. The patients from the erector spinae plane block group reported higher pain intensity despite using, on average, twice the amount of opioid compared to both other groups.

Elsabeeny et al. found no difference between ESPB and SPB in breast cancer surgeries. The reported pain intensity was comparable in both groups. The main limitation of their study was the control group, where morphine was used instead of PVB; therefore, only the need for rescue ketorolac analgesia was measured instead of opioid requirements [20].

In the study by Mekhaeil et al., SAPB provided more effective postoperative analgesia in patients undergoing modified radical mastectomy, with lower pain scores, less peri operative analgesic consumption, and longer duration of analgesia than ESPB [21].

The study by Ahuja et al. failed to show statistically significant differences in postoperative analgesia between ESPB and SAPB, and only the percentage of patients requiring rescue analgesia was recorded instead of those requiring a PCA pump like in our study, which could have affected the accuracy of the comparison [22].

### 4.5. Limitations

Only the first pain assessment was conducted exactly one hour after the surgery, while the subsequent assessments were carried out at fixed times, which may affect the accuracy of our results. Furthermore, variations in surgeons and the durations of procedures were not considered during randomization.

## 5. Conclusions

Both the serratus anterior plane block and the erector spinae plane block were as effective as the paravertebral block in achieving intraoperative analgesia.The serratus anterior plane block was equally as effective as the paravertebral block in achieving postoperative analgesia.The erector spinae plane block was significantly less effective in achieving postoperative analgesia than both the paravertebral block and serratus anterior plane block.

## Figures and Tables

**Figure 1 jcm-13-04836-f001:**
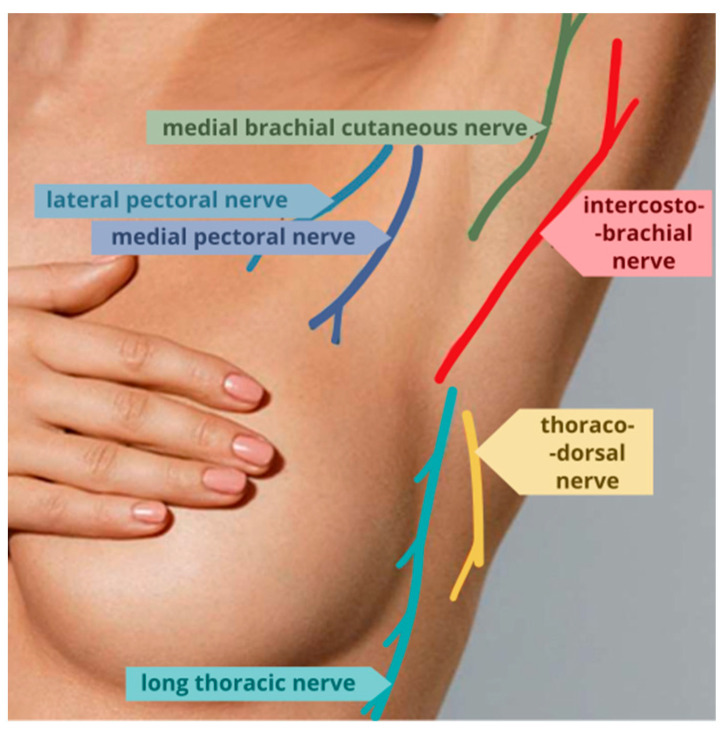
Innervation of the skin and subcutaneous tissue of the axillary line originating from the brachial plexus along with the intercostobrachial nerve originating from intercostal nerves Th2-3.

**Figure 2 jcm-13-04836-f002:**
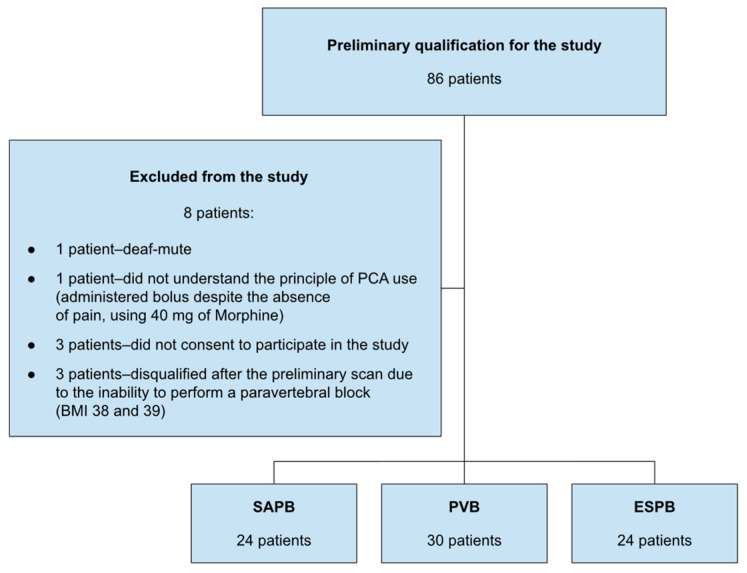
Qualification for the study.

**Figure 3 jcm-13-04836-f003:**
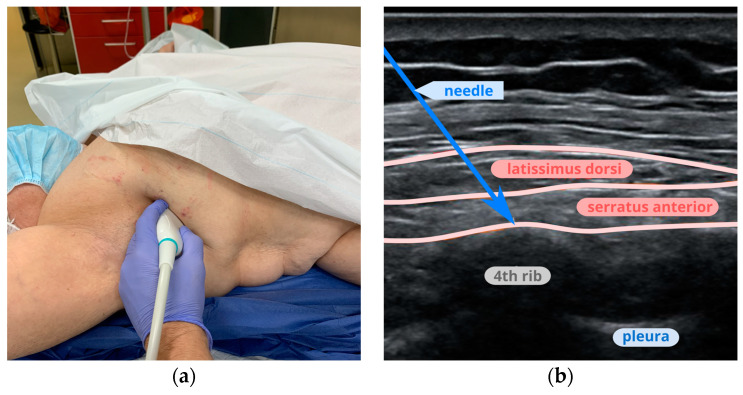
Serratus anterior plane block: (**a**) placement of the probe at the fourth-rib level for SAPB; (**b**) an ultrasound image of the serratus anterior and latissimus dorsi muscles.

**Figure 4 jcm-13-04836-f004:**
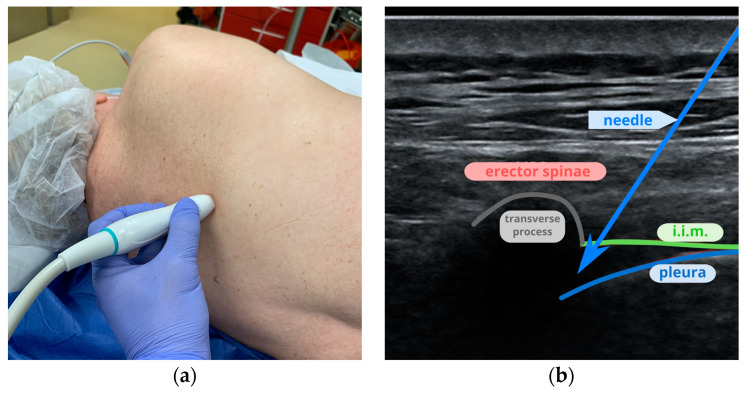
Paravertebral block: (**a**) placement of the probe between the fourth and fifth ribs for PVB; (**b**) an ultrasound image of the costotransverse joint and internal intercostal membrane (i.i.m.).

**Figure 5 jcm-13-04836-f005:**
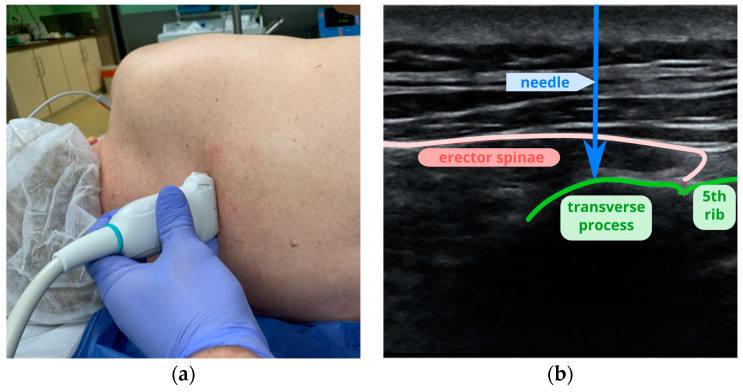
Erector spinae plane block: (**a**) placement of the probe at the fifth-rib level for ESPB; (**b**) a scan at the level of the costotransverse joint, where the probe is placed in a transverse projection; (**c**) a scan showing the transverse process, where the probe is placed in a longitudinal projection.

**Table 1 jcm-13-04836-t001:** Comparative characteristics of the studied groups (SAPB—serratus anterior plane block; PVB—paravertebral block; ESPB—erector spinae plane block) regarding the type of procedure.

	SAPB (*n* = 24)	PVB (*n* = 30)	ESPB (*n* = 24)	Total (*n* = 78)	*p*-Value
Procedure					0.8031 ^1^
Mastectomy + SNLD	4 (16.7%)	5 (17.9%)	3 (12.5%)	12 (15.8%)	
BCT + SNLD	16 (66.7%)	16 (57.1%)	15 (62.5%)	47 (61.8%)	
Madens’ mastectomy	2 (8.3%)	4 (14.3%)	4 (16.7%)	10 (13.2%)	
Quadrantectomy + SNLD	0 (0.0%)	1 (3.6%)	0 (0.0%)	1 (1.3%)	
Tumorectomy	2 (8.3%)	0 (0.0%)	1 (4.2%)	3 (3.9%)	
Quadrantectomy	0 (0.0%)	1 (3.6%)	0 (0.0%)	1 (1.3%)	
BCT	0 (0.0%)	1 (3.6%)	1 (4.2%)	2 (2.6%)	

^1^ Chi-square.

**Table 2 jcm-13-04836-t002:** Comparative characteristics of the studied groups (SAPB—serratus anterior plane block; PVB—paravertebral block; ESPB—erector spinae plane block) regarding intraoperative fentanyl consumption and postoperative morphine consumption.

	SAPB (*n* = 24)	PVB (*n* = 30)	ESPB (*n* = 24)	Total (*n* = 78)	*p*-Value
Fentanyl (mg)					0.4246 ^1^
Mean (SD)	0.12 (0.04)	0.12 (0.05)	0.11 (0.03)	0.12 (0.04)	
Range	0.10–0.20	0.10–0.30	0.10–0.20	0.10–0.30	
Median	0.10	0.10	0.10	0.10	
95% CI	[0.10;0.14]	[0.10;0.14]	[0.10;0.12]	[0.11;0.13]	
Morphine (mg)					0.0004 ^1^
Mean (SD)	5.4 (3.7)	4.4 (2.1)	9.4 (5.5)	6.3 (4.4)	^a^ 0.0074 ^2^
Range	1.0–15.0	1.0–18.0	3.0–23.0	1.0–23.0	^b^ 0.0005 ^2^
Median	4.0	4.0	8.0	5.0	
95% CI	[3.8;6.9]	[3.6;5.2]	[7.1;11.7]	[5.3;7.3]	

^1^ ANOVA Kruskal–Wallis; ^2^ post hoc tests; ^a^ SAPB vs ESPB; ^b^ PVB vs ESPB.

**Table 3 jcm-13-04836-t003:** Comparative characteristics of the studied groups (SAPB—serratus anterior plane block; PVB—paravertebral block; ESPB—erector spinae plane block) regarding pain intensity at rest.

	SAPB (*n* = 24)	PVB (*n* = 30)	ESPB (*n* = 24)	Total (*n* = 78)	*p*-Value
1 h after the procedure	*n* = 20	*n* = 28	*n* = 23	*n* = 71	0.8651 ^1^
mean (SD)	4.1 (1.8)	4.0 (1.7)	3.9 (1.6)	4.0 (1.6)	
median (range)	4.0 (2.0–9.0)	4.0 (1.0–7.0)	3.0 (2.0–8.0)	4.0 (1.0–9.0)	
95%CI	[3.3;4.9]	[3.3;4.7]	[3.2;4.6]	[3.6;4.4]	
4:00 PM	*n* = 24	*n* = 27	*n* = 24	*n* = 75	0.0017 ^1^
mean (SD)	2.6 (0.8)	2.6 (0.9)	3.6 (1.2)	2.9 (1.1)	^a^ 0.0129 ^2^
median (range)	2.0 (2.0–5.0)	2.0 (1.0–5.0)	4.0 (1.0–5.0)	3.0 (1.0–5.0)	^b^ 0.0071 ^2^
95%CI	[2.3;3.0]	[2.2;2.9]	[3.1;4.1]	[2.7;3.2]	
8:00 PM	*n* = 24	*n* = 27	*n* = 24	*n* = 75	0.0017 ^1^
mean (SD)	2.6 (0.8)	2.6 (0.9)	3.6 (1.2)	2.9 (1.1)	^a^ 0.0129 ^2^
median (range)	2.0 (2.0–5.0)	2.0 (1.0–5.0)	4.0 (1.0–5.0)	3.0 (1.0–5.0)	^b^ 0.0071 ^2^
95%CI	[2.3;3.0]	[2.2;2.9]	[3.1;4.1]	[2.7;3.2]	
12:00 AM	*n* = 2	*n* = 7	*n* = 6	*n* = 15	--
mean (SD)	2.5 (0.7)	2.4 (0.5)	2.8 (1.2)	2.6 (0.8)	
median (range)	2.5 (2.0–3.0)	2.0 (2.0–3.0)	2.5 (2.0–5.0)	2.0 (2.0–5.0)	
95%CI	[-3.9;8.9]	[1.9;2.9]	[1.6;4.1]	[2.1;3.1]	
4:00 AM	*n* = 0	*n* = 1	*n* = 1	*n* = 2	--
mean (SD)		2.0 (0.0)	1.0 (0.0)	1.5 (0.7)	
median (range)		2.0 (2.0–2.0)	1.0 (1.0–1.0)	1.5 (1.0–2.0)	
95%CI		[0.0;0.0]	[0.0;0.0]	[0.0;0.0]	
8:00 AM	*n* =24	*n* = 28	*n* = 24	*n* = 76	0.0176 ^1^
mean (SD)	2.1 (0.7)	2.0 (0.7)	2.7 (1.0)	2.3 (0.9)	^a^ 0.0388 ^2^
median (range)	2.0 (1.0–4.0)	2.0 (1.0–4.0)	3.0 (1.0–4.0)	2.0 (1.0–4.0)	
95%CI	[1.8;2.4]	[1.7;2.3]	[2.3;3.1]	[2.1;2.5]	
12:00 PM	*n* = 24	*n* = 29	*n* = 24	*n* = 77	>0.05 ^1^
mean (SD)	1.8 (0.5)	1.9 (0.8)	2.4 (1.0)	2.0 (0.8)	
median (range)	2.0 (1.0–3.0)	2.0 (1.0–4.0)	2.5 (1.0–4.0)	2.0 (1.0–4.0)	
95%CI	[1.6;2.0]	[1.6;2.2]	[2.0;2.8]	[1.8;2.2]	
4:00 PM	*n* = 24	*n* = 29	*n* = 24	*n* = 77	0.2182 ^1^
mean (SD)	1.8 (0.6)	1.8 (0.8)	2.2 (0.9)	1.9 (0.8)	
median (range)	2.0 (1.0–3.0)	2.0 (1.0–4.0)	2.0 (1.0–4.0)	2.0 (1.0–4.0)	
95%CI	[1.5;2.0]	[1.5;2.1]	[1.8;2.5]	[1.8;2.1]	

^1^ ANOVA Kruskal–Wallis; ^2^ post hoc tests; ^a^ SAPB vs ESPB; ^b^ PVB vs ESPB.

**Table 4 jcm-13-04836-t004:** Comparative characteristics of the studied groups (SAPB—serratus plane block; PVB—paravertebral block; ESPB—erector spinae plane block) regarding pain intensity during coughing.

	SAPB (*n* = 24)	PVB (*n* = 30)	ESPB (*n* = 24)	Total (*n* = 78)	*p*-Value
1 h after the procedure	*n* = 20	*n* = 28	*n* = 23	*n* = 71	0.8809 ^1^
mean (SD)	4.8 (1.6)	4.8 (1.7)	4.8 (1.4)	4.8 (1.6)	
median (range)	5.0 (2.0–8.0)	5.0 (1.0–8.0)	4.0 (3.0–8.0)	5.0 (1.0–8.0)	
95%CI	[4.1;5.5]	[4.2;5.5]	[4.2;5.4]	[4.4;5.2]	
4:00 PM	*n* = 24	*n* = 28	*n* = 24	*n* = 76	0.0010 ^1^
mean (SD)	3.5 (0.9)	3.5 (0.9)	4.5 (1.1)	3.8 (1.0)	^a^ 0.0080 ^2^
median (range)	3.0 (1.0–5.0)	3.0 (2.0–5.0)	4.5 (2.0–6.0)	4.0 (1.0–6.0)	^b^ 0.0045 ^2^
95%CI	[3.1;3.9]	[3.2;3.8]	[4.0;4.9]	[3.5;4.0]	
8:00 PM	*n* = 24	*n* = 28	*n* = 24	*n* = 76	0.0010 ^1^
mean (SD)	3.5 (0.9)	3.5 (0.9)	4.5 (1.1)	3.8 (1.0)	^a^ 0.0080 ^2^
median (range)	3.0 (1.0–5.0)	3.0 (2.0–5.0)	4.5 (2.0–6.0)	4.0 (1.0–6.0)	^b^ 0.0045 ^2^
95%CI	[3.1;3.9]	[3.2;3.8]	[4.0;4.9]	[3.5;4.0]	
12:00 AM	*n* = 2	*n* = 7	*n* = 6	*n* = 15	--
mean (SD)	3.5 (0.7)	3.6 (0.8)	3.7 (0.8)	3.6 (0.7)	
median (range)	3.5 (3.0–4.0)	3.0 (3.0–5.0)	3.5 (3.0–5.0)	3.0 (3.0–5.0)	
95%CI	[−2.9;9.9]	[2.8;4.3]	[2.8;4.5]	[3.2;4.0]	
4:00 AM	*n* = 0	*n* = 1	*n* = 1	*n* = 2	--
mean (SD)		3.0 (0.0)	1.0 (0.0)	2.0 (1.4)	
median (range)		3.0 (3.0–3.0)	1.0 (1.0–1.0)	2.0 (1.0–3.0)	
95%CI		[0.0;0.0]	[0.0;0.0]	[0.0;0.0]	
8:00 AM	*n* =24	*n* = 28	*n* = 24	*n* = 76	0.0128 ^1^
mean (SD)	3.1 (0.7)	3.1 (0.8)	3.7 (0.9)	3.3 (0.8)	^a^ 0.0478 ^2^
median (range)	3.0 (2.0–5.0)	3.0 (2.0–5.0)	4.0 (2.0–5.0)	3.0 (2.0–5.0)	
95%CI	[2.8;3.4]	[2.8;3.4]	[3.3;4.1]	[3.1;3.5]	
12:00 PM	*n* = 24	*n* = 29	*n* = 24	*n* = 77	0.0097 ^1^
mean (SD)	2.8 (0.5)	2.8 (0.8)	3.4 (0.9)	3.0 (0.8)	^a^ 0.0253 ^2^
median (range)	3.0 (2.0–4.0)	3.0 (2.0–5.0)	3.0 (2.0–5.0)	3.0 (2.0–5.0)	
95%CI	[2.6;3.0]	[2.5;3.1]	[3.0;3.7]	[2.8;3.1]	
4:00 PM	*n* = 24	*n* = 29	*n* = 24	*n* = 77	>0.05 ^1^
mean (SD)	2.8 (0.6)	2.6 (0.7)	3.2 (0.9)	2.8 (0.8)	
median (range)	3.0 (2.0–4.0)	3.0 (2.0–5.0)	3.0 (2.0–5.0)	3.0 (2.0–5.0)	
95%CI	[2.5;3.0]	[2.3;2.9]	[2.8;3.5]	[2.7;3.0]	

^1^ ANOVA Kruskal–Wallis; ^2^ post hoc tests; ^a^ SAPB vs ESPB; ^b^ PVB vs ESPB.

## Data Availability

The data presented in this study are available on request from the corresponding author.

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
