# Peer review of "A Comparison of the Effectiveness of the Serratus Anterior Plane Block and Erector Spinae Plane Block to that of the Paravertebral Block in the Surgical Treatment of Breast Cancer—A Randomized, Prospective, Single-Blinded Study"

_jcm, 2024, doi:10.3390/jcm13164836_

Round 1

Reviewer 1 Report

Comments and Suggestions for Authors

Dear authors of the manuscript,

The goal of this review is to improve the quality of your manuscript.

The manuscript is written on 14 pages. It contains all necessary parts of the manuscript: structured abstract, introduction, material and methods, results, discussion, conclusions and references. 

The title of the manuscript is clearly formulated.

Abstract: structured abstract. It contains all parts of a structured abstract. Total number of words in abstract is 377 (including words: Material and methods, Results, Conclusion’s; Keywords are not included). There are 5 Key words; one of them is according to the MeSH terminology.

1.      You missed to write Background/Objectives on the beginning of the Abstract

2.      Please DELETE citation from the Abstract (line 13). It is very unusual to have references citated in abstract

3.      Please remove (delete) keywords numbering

4.      You stated: “Paravertebral blockade (PVB) is a well-studied, effective method for brest surgery …” EFFECTIVE METHOD OF WHAT? PAIN CONTROL / MANAGEMENT, INTRAOPERATIVE ANAESTHESIA / ANALGESIA?

5.      Delete word “Objective” in the line 18 and instead in line 13 please write “Background/Objectives)

6.      NRS? You use the abbreviation for the first time in the text! Please write full name of the pain assessment tool (scale)

Introduction: The introduction provides an insight into the research issues.

1.      In line 53: How it is possible to number 4 from Reference list be followed by number 23? Please explain.

2.      Lines 59-60: “There are many studies in the literature comparing the effectiveness of the PVB to the ESPB or to the ESPB.” Which studies? On which references do you recall?

Methodology

1.      Line 76 and 77: “To assess the impact of SAPB, ESPB, and PVB blocks on the amount of morphine administered to patients.” WHEN? POSTOPERATIVELY?

2.      Line 79 and 80: “first day post-surgery six times daily from the first hour post-operation until the end of the second postoperative day”.

As I can see in the results (Table 3), the time when you measured postoperative pain was as following: 1 hour after the procedure, 4 PM, 8 PM, 12 AM, 4 AM, 8 AM, 12 PM, 4 PM. Am I right? It is so, what do you think, could the time when the patient went out from the operating room effect on the intensity of postoperative pain? For example, some patients went out OP room at for example 11 AM and some at 12:30 PM. And for both of these two patients you measured pain intensity at 4 PM? If it is so, from the point of methodology it isn’t good. And you have to write in the methodology very precisely the pain assessment. time.

3.      How did you calculate the sample size? In statistic, there is exact methodology how to calculate the sample size which is necessary for adequate statistical data analysis.

4.      How did you perform randomization of the patients? Which computer-based table assigned patients into groups? You must state it precisely.

5.      In the lines 87-88 you don’t mention the use of 1% lidocaine for infiltrative anesthesia.

6.      Line 137: general INHALATION anesthesia OR general BALANCED anesthesia?

7.      Line 138: PREMEDICATION or CO-INDUCTION of anesthesia?

8.      Did all patients receive 2 mg of midazolam and 0.1 mg of fentanyl for "premedication", regardless of whether they weighed 65 kg or 80 kg?

9.      Line 140-141: How much rocuronium? 0,6 or 1,0 or 1,2 mg/kg?

10.   In the methodology I can’t see any word about: type of the surgery (surgical technique), duration of the surgery, did all surgeries were performed by the same surgeon?

According to the Anesthesiologist’s Manual of Surgical Procedures 6th edition (Editors: Richard A. Jaffe, Clifford A. Schmiesing, Brenda Golianu) there is difference between pain SCORES among different surgical procedures. Also, there is differences in SURGICAL TIME. This might have influence on postoperative pain

11.   How did you evaluate:To evaluate the safety of the paravertebral block and the applied fascial blocks SAPB and ESPB.”?

12.   Please specify in “Methodology” which statistical methods did you use and for which variables.

13.   Please pay attention on yours following sentence: “For the times 12:00 AM and 4:00 AM, statistical tests could not be calculated due to small or absent sample sizes.” NOW YOU CAN UNDERSTAND MY QUESTION ABOUT SAMPLE SIZE

14.   In the Results you are explaining how did you assess pain in patients during the cough???

15.   How do you explain sentence “There were statistically significant differences in pain intensity at rest at 4:00 PM with respect to the type of block used”

Discussion and conclusions:

16.   Lines 314-317: Reference 20 doesn’t belong to Elsabeeny et al.

17.   Lines 318-320: Reference 21 doesn’t belong to Mekhaeil et al.

18.   Lines 321-324: Reference 22 doesn’t belong to Ahuja et al.

19.   The conclusions do not derive from the research objectives (please see the material and methods!)

20.   Does your study have any limitations?

References: There are 23 references. 15 references are from 2020 onwards.

PLEASE correct the References according to the Reference List and Citations Style Guide for MDPI Journals. https://mdpi-res.com/data/mdpi_references_guide_v5.pdf

Comments on the Quality of English Language

Proofreading of the manuscript is necessary. 

Author Response

in abstract is 377 (including words: Material and methods, Results, Conclusion’s; Keywords are not included). There are 5 Key words; one of them is according to the MeSH terminology.

Thank you for pointing out the length of the abstract, we have shortened it to 267 words

  1.     You missed to write Background/Objectives on the beginning of the Abstract

Agree, we have changed it to meet the standard abstract form.

  1.     Please DELETE citation from the Abstract (line 13). It is very unusual to have references citated in abstract

Thank you for the suggestion, we have deleted it.

  1.     Please remove (delete) keywords numbering

Of course, we have deleted it to meet the standard abstract form.

  1.     You stated: “Paravertebral blockade (PVB) is a well-studied, effective method for brest surgery …” EFFECTIVE METHOD OF WHAT? PAIN CONTROL / MANAGEMENT, INTRAOPERATIVE ANAESTHESIA / ANALGESIA?

 Agree, changed to “effective method of analgesia for breast surgery”

  1.     Delete word “Objective” in the line 18 and instead in line 13 please write “Background/Objectives)

Agree, changed.

  1.     NRS? You use the abbreviation for the first time in the text! Please write full name of the pain assessment tool (scale)

Agree, changed.

Introduction: The introduction provides an insight into the research issues.

  1.     In line 53: How it is possible to number 4 from Reference list be followed by number 23? Please explain.

Agree, deleted.

  1.     Lines 59-60: “There are many studies in the literature comparing the effectiveness of the PVB to the ESPB or to the ESPB.” Which studies? On which references do you recall?

Multiple studies are mentioned in the Discussion; therefore, we do not see the need to reference them in the introduction.

Methodology

  1.     Line 76 and 77: “To assess the impact of SAPB, ESPB, and PVB blocks on the amount of morphine administered to patients.” WHEN? POSTOPERATIVELY?

Agree, we clarified.

  1.     Line 79 and 80: “first day post-surgery six times daily from the first hour post-operation until the end of the second postoperative day”

As I can see in the results (Table 3), the time when you measured postoperative pain was as following: 1 hour after the procedure, 4 PM, 8 PM, 12 AM, 4 AM, 8 AM, 12 PM, 4 PM. Am I right? It is so, what do you think, could the time when the patient went out from the operating room effect on the intensity of postoperative pain? For example, some patients went out OP room at for example 11 AM and some at 12:30 PM. And for both of these two patients you measured pain intensity at 4 PM? If it is so, from the point of methodology it isn’t good. And you have to write in the methodology very precisely the pain assessment. time.

Despite a gap in pain measurements for the earliest procedures, the accuracy was sufficient to demonstrate statistically significant differences between the procedures. This is confirmed by overall morphine consumption, which correlates with the pain scores and is not affected by the aforementioned limitation. We have identified this as one of the study's limitations

  1.     How did you calculate the sample size? In statistic, there is exact methodology how to calculate the sample size which is necessary for adequate statistical data analysis.

We acknowledge that the study is underpowered. Although the initial design was different, we encountered unexpected difficulties due to the pandemic. We have identified the sample size as a major limitation.

How did you perform randomization of the patients? Which computer-based table assigned patients into groups? You must state it precisely.

The computer-based table used to assign patients into groups was generated using a randomization table created in Microsoft Excel. We added this information

  1.     In the lines 87-88 you don’t mention the use of 1% lidocaine for infiltrative anesthesia.

We think it is an unnecessary repetition.

  1.     Line 137: general INHALATION anesthesia OR general BALANCED anesthesia?

Agree, corrected.

  1.     Line 138: PREMEDICATION or CO-INDUCTION of anesthesia?

Agree, corrected.

  1.     Did all patients receive 2 mg of midazolam and 0.1 mg of fentanyl for "premedication", regardless of whether they weighed 65 kg or 80 kg?

Yes.

  1.     Line 140-141: How much rocuronium? 0,6 or 1,0 or 1,2 mg/kg?

0,6 mg/kg, we added this information

  1.   In the methodology I can’t see any word about: type of the surgery (surgical technique), duration of the surgery, did all surgeries were performed by the same surgeon?

According to the Anesthesiologist’s Manual of Surgical Procedures 6th edition (Editors: Richard A. Jaffe, Clifford A. Schmiesing, Brenda Golianu) there is difference between pain SCORES among different surgical procedures. Also, there is differences in SURGICAL TIME. This might have influence on postoperative pain.

Table 1. is all about types of surgeries, and the usual place for this type of data is the results section. We acknowledge variations in surgeons and durations of procedures were not considered during randomization and included it the limitations of the study.

.11.   How did you evaluate:To evaluate the safety of the paravertebral block and the applied fascial blocks SAPB and ESPB.”?

There was a paragraph about it in the original manuscript although we separated it as a subsection 2.6.

  1.   Please specify in “Methodology” which statistical methods did you use and for which variables.

Methods are described in the results section in the tables 2, 3 and 4.

  1.   Please pay attention on yours following sentence: “For the times 12:00 AM and 4:00 AM, statistical tests could not be calculated due to small or absent sample sizes.” NOW YOU CAN UNDERSTAND MY QUESTION ABOUT SAMPLE SIZE

The issue is not related to the group size but to the lack of measurements during the night due to the patients' need for sleep.

  1.   In the Results you are explaining how did you assess pain in patients during the cough???

What do mean by this statement? In our opinion pain assessment during cough is very important after this type of the operation.

  1.   How do you explain sentence “There were statistically significant differences in pain intensity at rest at 4:00 PM with respect to the type of block used”

Corrected for: “At 4:00 PM, there were significant differences in pain levels at rest depending on the type of anesthesia block used.”

Discussion and conclusions:

  1.   Lines 314-317: Reference 20 doesn’t belong to Elsabeeny et al.

Agree, corrected

  1.   Lines 318-320: Reference 21 doesn’t belong to Mekhaeil et al.

Agree, corrected

  1.   Lines 321-324: Reference 22 doesn’t belong to Ahuja et al.

Agree, corrected

  1.   The conclusions do not derive from the research objectives (please see the material and methods!)

Could you please clarify what you mean by this statement and provide more details?

  1.   Does your study have any limitations?

We added a section about limitations.

References: There are 23 references. 15 references are from 2020 onwards.

PLEASE correct the References according to the Reference List and Citations Style Guide for MDPI Journals. https://mdpi-res.com/data/mdpi_references_guide_v5.pdf

Agree, corrected.

Reviewer 2 Report

Comments and Suggestions for Authors

The Study Seems to be well performed and is adequately written.

I have only Minor commenrs:

in the abstract conclusions the authors State:

“The EPSB blockade proved to be as efecctive as the SAB…“

this may be true for intraoperative Fentanyl consumption. In general I think EPSB is less effective than the other methods. This should be written more pre isely.

authors should to my opinion included a discussion of adverse Effects. If EPSB is much safer than the other methods, it could make some sense to Apply this method, although it may be less effective.

3. please add limitations as there are numerous o es (Control Group, sample size, etc.)

Author Response

“The EPSB blockade proved to be as effective as the SAB…“

this may be true for intraoperative Fentanyl consumption. In general I think EPSB is less effective than the other methods. This should be written more pre isely.6

While this may be true we do not have direct evidence for it, at least in this study.

  1. authors should to my opinion included a discussion of adverse Effects. If EPSB is much safer than the other methods, it could make some sense to Apply this method, although it may be less effective.

We do not discuss adverse effect because in our study there was none.

  1. please add limitations as there are numerous o es (Control Group, sample size, etc.)

We have added the section about limitations.

Round 2

Reviewer 1 Report

Comments and Suggestions for Authors

Dear Authors of the manuscript,

11.     I still did not get the answer about the sample size. I still dont know how you calculated sample number (unless I rule out your diplomatic response: “We acknowledge that the study is underpowered. … We have identified the sample size as a major limitation …).

22.     Reviewer: Did all patients receive 2 mg of midazolam and 0.1 mg of fentanyl for "premedication", regardless of whether they weighed 65 kg or 80 kg?

Authors: Yes.

Sorry, I have to say this, but this is not quite right.

OXFORD HANDBOOK OF ANAESTHESIA FIFTH EDITION (p.407-408):

Opioids work synergistically with propofol to allow a dose reduction, as

well as obtund the autonomic response to airway instrumentation.

• Timing of administration should take into consideration time to peak

effect. Fentanyl (1– 2 micrograms/ kg) should be administered 3– 5min

prior to the induction agent, whereas alfentanil (10– 30 micrograms/ kg)

or remifentanil can be administered concurrently due to their faster

onset of action.

Adjuncts

• Midazolam (0.5– 5mg) is commonly used as a co- induction agent

to reduce the propofol dose required, therefore minimising the

cardiovascular effects of propofol. It can also reduce the incidence

of PONV.

33.     Reviewer: In the Results you are explaining how did you assess pain in patients during the cough???

Authors: What do mean by this statement? In our opinion pain assessment during cough is very important after this type of the operation.

Yes, I agree that pain assessment during cough is very important after this type of surgery as well as after cardiac surgery, general and visceral surgery …. My question is very clear: when the patient was coughing how did you assess the pain? Did you use any scale? Which scale did you use?

44.     Too many limiting factors of the study that may contribute to the misinterpretation of the obtained results.

Comments on the Quality of English Language

Proofreading of the manuscript in English is required.

Author Response

Thank you for the thorough and insightful analysis of our manuscript. Your input is invaluable in enhancing its scientific merit.

I still did not get the answer about the sample size. I still don’t know how you calculated sample number (unless I rule out your diplomatic response: “We acknowledge that the study is underpowered. … We have identified the sample size as a major limitation …).

Thank you for bringing this issue to our attention. We have recalculated the sample size and power of the study and are pleased to report that, despite challenges with data collection, our study is not underpowered. We have added a new subsection, 2.8: Sample Size and Power of the Study.

  1.     Reviewer: Did all patients receive 2 mg of midazolam and 0.1 mg of fentanyl for "premedication", regardless of whether they weighed 65 kg or 80 kg?

Authors: Yes.

Sorry, I have to say this, but this is very scientifically and professionally wrong.

OXFORD HANDBOOK OF ANAESTHESIA FIFTH EDITION (p.407-408):

Opioids work synergistically with propofol to allow a dose reduction, as

well as obtund the autonomic response to airway instrumentation.

  • Timing of administration should take into consideration time to peak

effect. Fentanyl (1– 2 micrograms/ kg) should be administered 3– 5min

prior to the induction agent, whereas alfentanil (10– 30 micrograms/ kg)

or remifentanil can be administered concurrently due to their faster

onset of action.

Adjuncts

  • Midazolam (0.5– 5mg) is commonly used as a co- induction agent

to reduce the propofol dose required, therefore minimising the

cardiovascular effects of propofol. It can also reduce the incidence

of PONV.

The objective of this particular study is to assess the effectiveness of regional blockade, which is unrelated to induction and intubation. While these factors may have a minimal influence on the results, we did not find any differences in intraoperative effectiveness. It would be very surprising to find any significant impact of such small differences in induction dose on postoperative analgesia.

We understand your concerns about the overall quality of our procedures, but we adhere to a local induction protocol that has been proven to be effective and safe in the past. While the instructions in the manuals you cite are valuable, they are not strict guidelines, as there are no proven exact doses of induction medications. If you are aware of such evidence, we would be happy to make an effort to revise our local protocol and apply it in future studies.

  1.     Reviewer: In the Results you are explaining how did you assess pain in patients during the cough???

Authors: What do mean by this statement? In our opinion pain assessment during cough is very important after this type of the operation.

Yes, I agree that pain assessment during cough is very important after this type of surgery as well as after cardiac surgery, general and visceral surgery …. My question is very clear: when the patient was coughing how did you assess the pain? Did you use any scale? Which scale did you use?

We acknowledge that our description may be unclear to some individuals. Therefore, we have revised subsection 2.5, "Postoperative Pain Intensity Measurement," to provide further clarification.

  1.     Too many limiting factors of the study that may contribute to the misinterpretation of the obtained results.

No study is without limitations. However, after analyzing our study and comparing it with similar research in the field, we conclude that it provides valuable and important contributions.